# Introduction to the Class of Prefractal Graphs

**Rasul Kochkarov [1,*]** and **Azret Kochkarov [1,2]**

1. Department of Data Analysis and Machine Learning, Faculty of Information Technology and Big Data Analysis, Financial University under the Government of the Russian Federation, Leningradsky Prospekt 49/2, 125167 Moscow, Russia; akochkar@gmail.com
2. Moscow Aviation Institute, National Research University, Volokolamskoe shosse 4, 125993 Moscow, Russia
* Correspondence: rasul_kochkarov@mail.ru

**Abstract:** Fractals are already firmly rooted in modern science. Research continues on the fractal properties of objects in physics, chemistry, biology and many other scientific fields. Fractal graphs as a discrete representation are used to model and describe the structure of various objects and processes, both natural and artificial. The paper proposes an introduction to prefractal graphs. The main definitions and notation are proposed—the concept of a seed, the operations of processing a seed, the procedure for generating a prefractal graph. Canonical (typical) and non-canonical (special) types of prefractal graphs are considered separately. Important characteristics are proposed and described—the preservation of adjacency of edges for different ranks in the trajectory. The definition of subgraph-seeds of different ranks is given separately. Rules for weighting a prefractal graph by natural numbers and intervals are proposed. Separately, the definition of a fractal graph as infinite is given, and the differences between the concepts of fractal and prefractal graphs are described. At the end of the work, already published works of the authors are proposed, indicating the main backlogs, as well as a list of directions for new research. This work is the beginning of a cycle of works on the study of the properties and characteristics of fractal and prefractal graphs.

**Keywords:** prefractal graph; noncanonical graph; generation procedure; trajectory

**MSC:** 05C51; 05C63; 05C75; 05C76; 68R10

## 1. Introduction

Fractals are recognized in the form of blood vessels [1], plants [2], coastlines [3], lightnings [4] and the structure of the Universe [5]. Recently, publications have also appeared on the fractality of complex networks, both of natural origin and artificial self-organizing networks [6]. This includes the Internet, social networks and biological networks [7–12].

Over the past decade, the study of fractal sets has become firmly established in science [13–15]. A great popularizer of fractal science is Benoit B. Mandelbrot; to understand the extent of the manifestation of the fractality of objects and processes, you should read his monograph [16]. Fractals are used to describe the structure of objects, and various methods are being developed for data processing and analysis [17].

The term fractal graph used in various publications has a similar meaning, but with some features. In [18], the representation of fractals in the form of line graphs. Such fractals as figures of Koch, Sierpinski, Minkowski, etc., are considered. For all figures, fractal dimensions are calculated, and an approach is proposed for grouping vertices into classes. In [13], based on the properties of fractals, a random sequence of hierarchical scale-free graphs is generated that has similar properties. In [19], a new algorithm for computing fractal dimension of rectifiable irregular graphs was proposed. Fractal graphs are being researched in different directions. The recognition of fractal graphs implies the definition of structures of natural or artificial objects as fractal. Separately, it should be noted the construction of fractal structures with specified initial parameters. Property

exploration includes the transfer of local properties to the global level or the emergence of new properties that are not inherent in local parts. Characteristics are considered both structural and numerical. Multicriteria optimization includes a description of a set of alternatives, problem definitions, development of algorithms, and evaluation of solutions.

In the works cited above, the term fractal graph was used mainly for finite graphs; in rare cases, infinite graphs were considered. Further, the concepts of a fractal graph as an infinite graph and a prefractal graph as a finite graph are separated, which corresponds to similar concepts of fractal and prefractal. Despite the closeness of the concepts of fractal and prefractal graphs, fractal graphs require additional approaches to study. Due to the emergence of new properties and characteristics in infinite objects. In this paper, prefractal (finite) graphs are considered; in one of the subsections, a brief definition of a fractal graph is given.

Introduction to the theory of fractal graphs implies familiarization with the class of fractal (prefractal) graphs. We propose a general approach to the description and construction of prefractal (fractal) graphs. In fact, we can talk about a separate class of graphs. There are a large number of publications where the subclasses of prefractal graphs are considered with their own terminology and definitions. Most of these graphs can be classified as canonical (typical) or noncanonical (special) prefractal graphs.

In the terminology of prefractal graphs [20–22], the families of self-similar graphs, such as Farey graphs, 2-dimensional Sierpiński gasket graphs, Hanoi graphs, modified Koch graphs, Apollonian graphs, pseudofractal scale-free webs, fractal scale-free networks, etc. [23–25], are noncanonical prefractal graphs.

On the other hand, prefractal graphs are also dynamic graphs [26–28]. Large prefractal graphs are used to build graph models with the help of which optimization problems are solved [29–33].

## 2. Basic Definitions and Notations

### 2.1. Operation of Replacing Vertex by Seed

To designate a graph, the generally accepted notation $G = (V, E)$ [34–37] is used. To define prefractal and fractal graphs, we give additional definitions. Other definitions are given in [38,39].

A *seed* $H = (W, Q)$ is a connected graph with unnumbered vertices $v \in W$. An addition of the graph vertex splitting operation is the vertex replacement by a seed (RVS) operation. The essence of this addition is that the split vertex is replaced by a seed $H$. The RVS operation is implemented as follows.

In $G = (V, E)$, the vertex $v_0 \in V$ intended for replacement has its environment of a vertex—the set $U$ of all vertices adjacent to the vertex $v_0$, and the set $R$ of all edges incident to the vertex $v_0 : R = \{r = (v_0, u) : u \in U\}$. Next, the mapping $\varphi$ of vertices $u \in U$ to the set of vertices of the seed $H$ is defined (see Figure 1a,b):

$$\varphi : U \to W, \tag{1}$$

that is, each vertex $u \in U$ is assigned a vertex $\varphi(u) = v \in W$ of $H = (W, Q)$.

After that, the ends of the edges $r = (v_0, u) \in R$ of the environment $v_0$ are replaced by the vertex $v = \varphi(u) \in W$ defined by mapping (1) (see Figure 1c,d).

The «old» edge $e = (v_0, u)$ in the modified form $(v, u)$ retains its original designation (numbering). The RVS operation is considered completed as soon as for each edge $(v_0, u) \in R$, $u \in U$, the vertex $v_0$ is replaced by the vertex $v = \varphi(u)$ of the seed $H$ according to the mapping (1). New unnumbered vertices are assigned numbers, taking into account the already existing numbers of other vertices of the given graph. Similarly, designations (numbers) are assigned to new edges that have replaced the «old» vertex. It should be noted separately that seeds are also called graphlets [40–43].

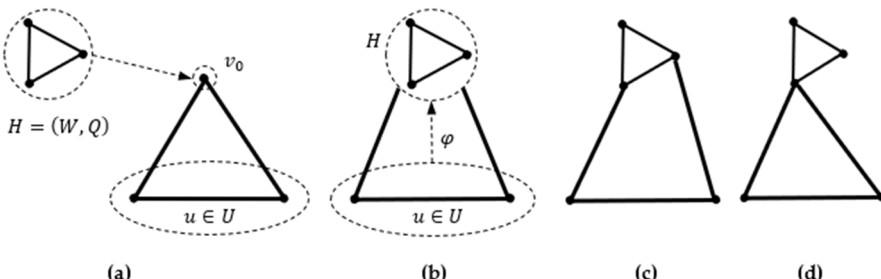

**Figure 1.** Operation of the replacement of vertex $v_0$ by seed $H$: (**a**) selection of a vertex for replacement and a replacement seed; (**b**) replacing a vertex and determining the environment of incident edges; (**c**) old edges are not adjacent; (**d**) the adjacency of the old edges is preserved.

### 2.2. Procedure for Generating Prefractal Graph

Let us consider a step-by-step process of constructing a prefractal graph and applying the RVS operation. At stage $l = 1$, the vertices and edges in the given seed $H = (W, Q)$ are numbered, and the resulting graph is denoted by $G_1 = (V_1, E_1)$. At stage $l = 2$, all vertices of $G_1$ are replaced by the seed $H$, the resulting graph is denoted by $G_2 = (V_2, E_2)$. Figure 2 shows the generation $G_1$ by seed $H$—a complete 3-vertex graph (triangle) with the arbitrary adjacency of «old» edges: (a) the vertices replaced by the seed are circled with small dashed circles; (b) the seeds replacing the vertices are circled by middle dashed circles; (c) the old edges of $G_2$ are marked with bold lines.

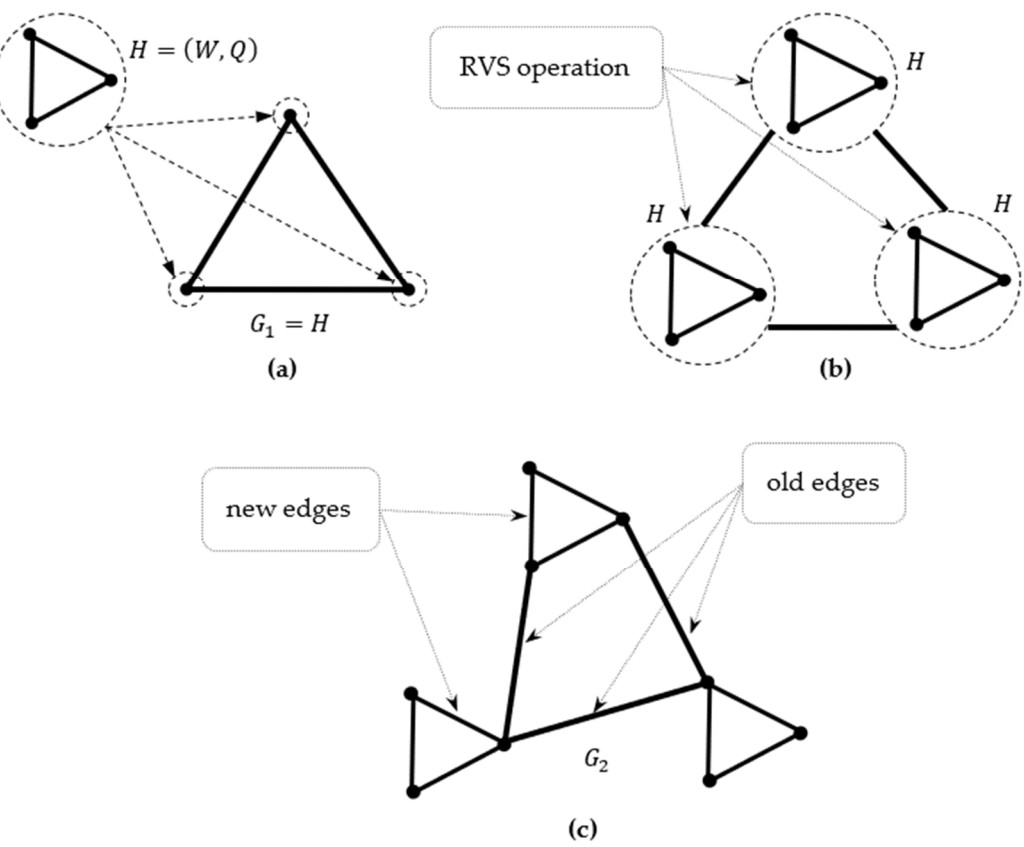

**Figure 2.** The generation $G_1$ by seed $H$: (**a**) selection of vertices for replacement and a replacement seed; (**b**) replacing vertices (RVS) and determining the environment of incident edges; (**c**) adjacency of old edges is arbitrary.

At each next stage $l = 3, 4, \ldots, L$, the operation RVS is applied to the vertices of the graph $G_{l-1}$. Upon completion of stage $L$, a graph $G_L = (V_L, E_L)$ is obtained, which is called prefractal. At each stage $l = 2, 3, \ldots, L$ of the RVS operation, $e \in E_{l-1}$ retain their

designation (numbering) and are called *old edges* in the trajectory $G_l, G_{l+1}, \ldots, G_L$. The edges of the replacing seeds are called *new edges* of the current graph $G_l$, and the set of new edges is $V_l \backslash V_{l-1}$. Thus, $G_{l+1}$ is obtained as a result of applying the RVS operation to each of the vertices of the set $V_l$.

$G_L = (V_L, E_L)$ is a *prefractal graph* with the set of vertices $V_L$, and the set of edges $E_L$. It is determined recursively, in $G_l = (V_l, E_l)$, $l = 1, 2, \ldots, L - 1$, each of its vertices is replaced by a seed $H = (W, Q)$. At stage $l = 1$, the prefractal graph $G_1$ corresponds to the seed $H$: $G_1 = H$. It is said about the described process that $G_L$ is generated by $H$. The sequence of prefractal graphs $G_1, G_2, \ldots, G_L$ is called a *trajectory*. $G_l$ is a prefractal graph of rank $l$. Edges of rank $L$ are new edges, and edges of rank $l$ are old edges. A simplified notation $G_L$ is used for $G_L = (V_L, E_L)$. Separately, there will be a description of the difference between canonical and noncanonical prefractal graphs.

Figure 3 shows the trajectory $G_1, G_2, G_3$ of the prefractal graph $G_3$ generated by a triangle $H$ with the arbitrary adjacency of the old edges.

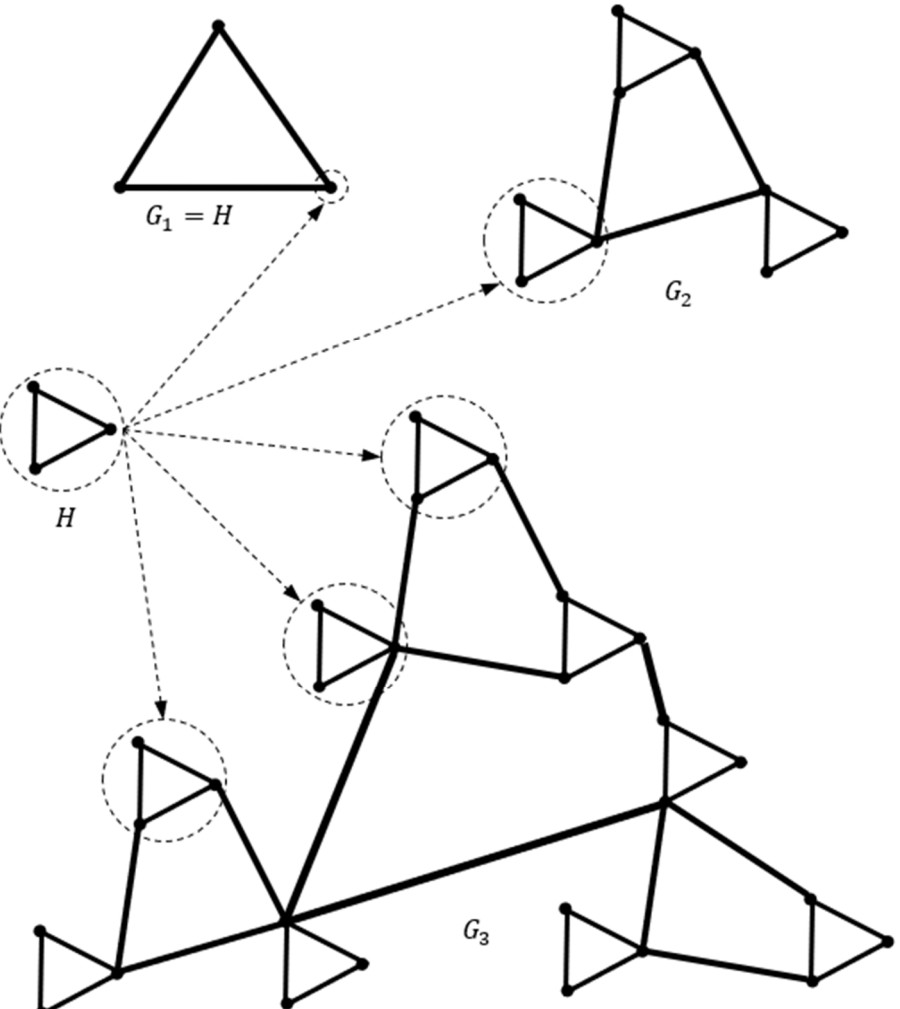

**Figure 3.** Trajectory $G_1, G_2, G_3$ of the prefractal graph $G_3$. The dotted circles indicate the seeds.

An addition of the process of generating $G_L$ is such a case when instead of $H$, a set $\mathrm{H} = \{H\} = \{H_1, H_2, \ldots, H_t, \ldots, H_T\}$, $T \geq 2$ is given. In the transition from $G_l$ to $G_{l+1}$ vertex is replaced by $H_t \in \mathrm{H}$. In accordance with this, one of the seeds $H_t$ is taken as the prefractal graph $G_1$. The cardinality of the set of vertices of the seed $H_t = (W_t, Q_t)$, that is, the number of vertices, is respectively equal to $|W_t| = n_t$. For simplification, the following notations are used: seed $H \in \{H_t\}$, where the number of vertices is $n = \max n_t$, and the number of edges is $q = \max q_t$.

Figure 4 shows $G_3$ generated by a set H with arbitrarily adjacency of old edges: (a) H $= \{H_1, H_2, H_3\}$, $H_1$ is a complete 3-vertex graph-triangle, $H_2$ is a complete 2-vertex graph, $H_3$ is a 4-vertex graph star; (b) the seeds that again replaced the vertices are outlined by small dashed circles; (c) large dashed circles outline the subgraphs that appeared instead of the graph vertices $G_1$ after the second stage of replacing the vertices with a seed.

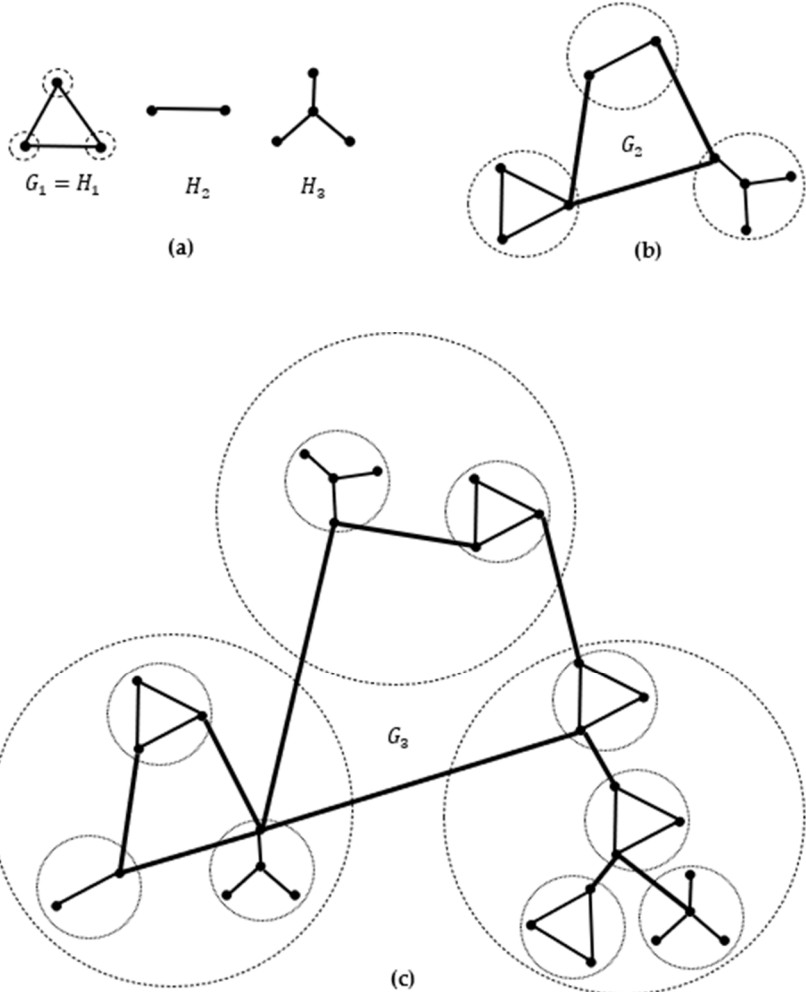

**Figure 4.** $G_3$ generated by a set H : (**a**) set of seeds $\{H_1, H_2, H_3\}$; (**b**) prefractal graph $G_2$; (**c**) prefractal graph $G_3$, small circles indicate new seeds, large circles indicate seeds of the second stage.

*2.3. Canonical and Noncanonical Prefractal Graphs*

A distinctive feature of the generating process is that at each $l = 2, 3, \ldots, L$ in the graph $G_{l-1} = (V_{l-1}, E_{l-1})$, each vertex $v \in V_{l-1}$ is replaced by a seed. Prefractal graphs resulting from such a process are called *canonical*.

A *noncanonical* prefractal graph is generated with one fundamental difference: when passing from $G_{l-1}$ to $G_l$ in the trajectory $G_1, G_2, \ldots, G_{l-1}, G_l, \ldots G_L$, not every vertex $v \in V_{l-1}$ of $G_{l-1}$ is replaced by H, but only a subset $V_{l-1}^* \subset V_{l-1}$.

A non-trivial case of the RVS operation is the replacement of a vertex not by a seed, but by a graph $G_{l-t}$, $t = 1, 2, \ldots, l-1$ from the trajectory $G_1, G_2, \ldots, G_{l-1}$. Figure 5 shows a noncanonical prefractal graph $G_4$, in which, at the last fourth stage of generation, two vertices are replaced by a graph $G_2$ from the trajectory $G_1, G_2, \ldots, G_3$.

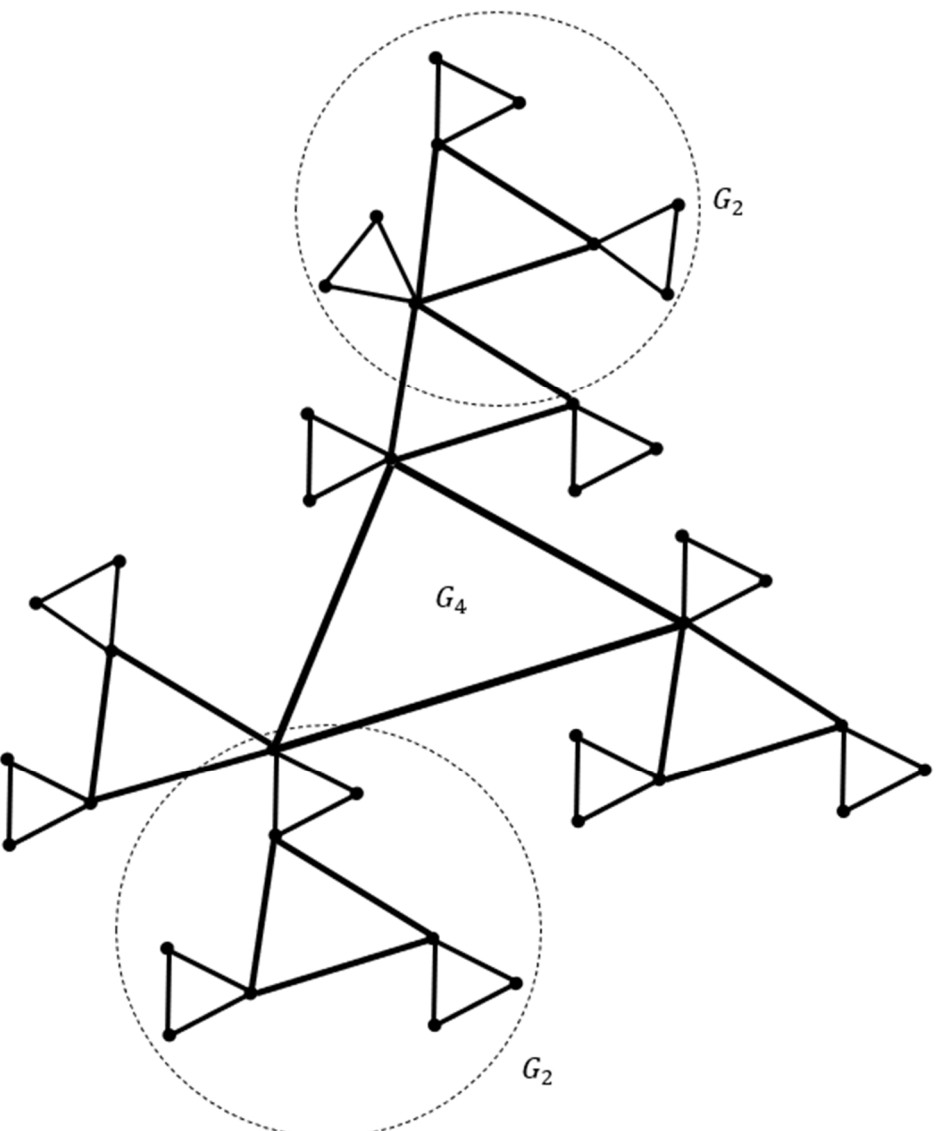

**Figure 5.** Noncanonical prefractal graph $G_4$ generated with $G_2$.

As mentioned in the introduction, graphs from the family of self-similar graphs can be defined as a prefractal noncanonical graph. Figure 6 shows noncanonical $G_3'$ (a) and canonical $G_3$ (b) prefractal graphs.

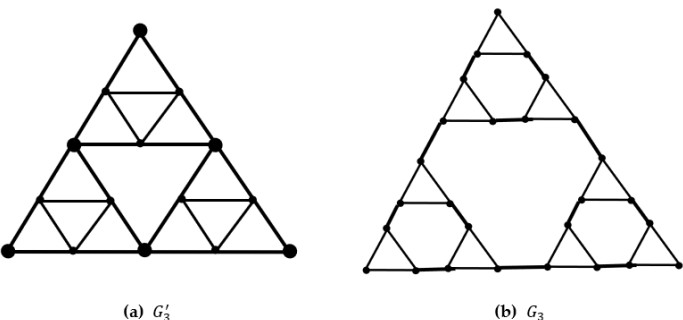

(a) $G_3'$        (b) $G_3$

**Figure 6.** (**a**) Noncanonical $G_3'$; (**b**) canonical $G_3$.

### 2.4. Prefractal Multigraphs

Not only ordinary graphs, but also multigraphs, including directed graphs, can be used as seeds. A multigraph (or pseudograph) is a graph that allows the presence of

multiple edges that have the same end vertices, that is, two vertices can be connected by more than one edge. There are several different ways to label the edges of a multigraph. In one case, an edge is defined by the vertices it connects, and each edge can be repeated multiple times. Otherwise, the edges are equal and must have their own identification. A prefractal graph generated by a set of seed multigraphs or a single seed multigraph is called a *prefractal multigraph*.

Figure 7 shows a prefractal multigraph $G_3$ generated by a set $H = \{H_1, H_2\}$ with the arbitrarily adjacency of old edges. The edges $e'$ and $e''$ are adjacent in $G_2$, but are no longer so in $G_3$.

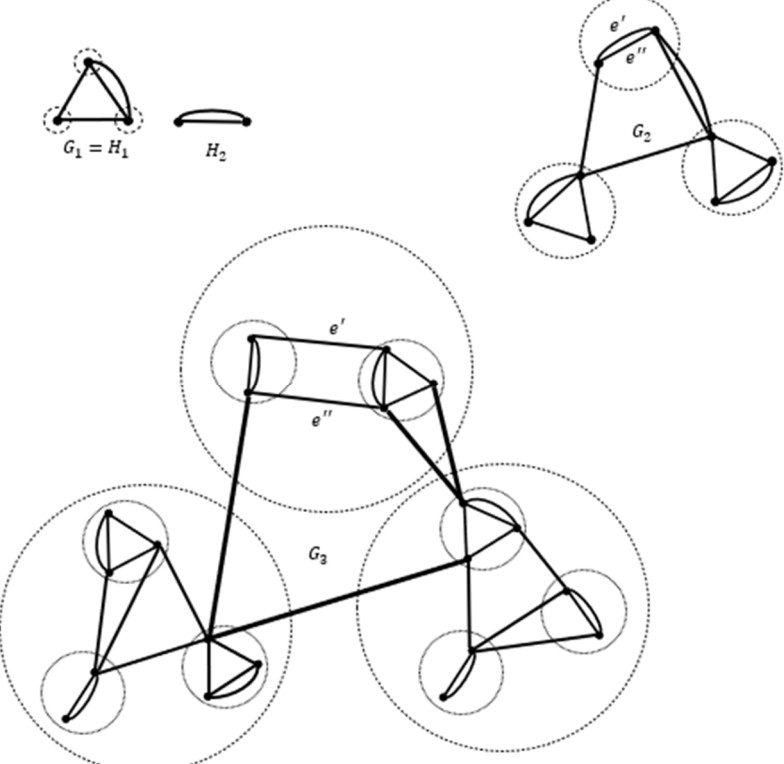

**Figure 7.** Multigraph $G_3$ generated by a set $H = \{H_1, H_2\}$.

Figure 8 shows a prefractal oriented multigraph $G_3$ generated by a set $H = \{H_1, H_2\}$ with the arbitrarily adjacency of old edges. The arc $e'$ of the seed $H_2$ is a loop. The incidences of arcs change: the arc $e''$ forming a loop in $G_2$ ceases to be such in $G_3$, the beginning of the arc leaves the vertex $v'$, and the end enters $v''$.

Prefractal graphs generated by hypergraphs are complicated objects and require a separate study. Recall that in a hypergraph, an edge can connect not only two vertices, but any subset of vertices.

### 2.5. Definitions and Characteristics of a Prefractal Graph

A prefractal graph $G_L$ is an $(n, q, L)$-*graph* if its $n$-vertex seed $H = (W, Q)$ has a set $Q$ consisting of $q = |Q| \le n(n-1)/2$ edges. $G_L$ is an $(n, L)$-*graph* if it is generated by a set of $n$-vertex connected seeds of the same degree. If the only seed $H$ constituting the set H is a complete $n$-vertex graph ($q = n(n-1)/2$), then the prefractal graph $G_L$ is $(n, L)$-*graph with complete seed*. The result of constructing a prefractal graph $G_L$ is a trajectory $G_1, G_2, \ldots, G_l, \ldots G_L$, where the parameter $l$ is the *rank* of $G_l$.

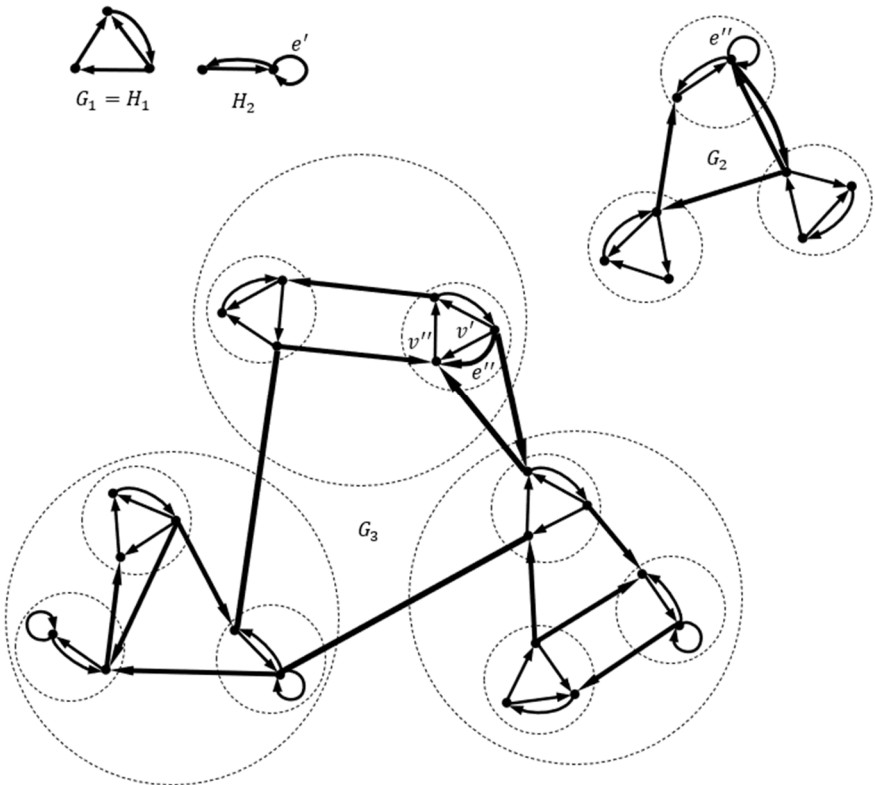

**Figure 8.** Oriented multigraph $G_3$.

Using the RVS operation in the process of generating $G_L$ for $G_l = (V_l, E_l), l \in \{1, 2, \ldots, L-1\}$ allows to enter a mapping $\varphi : V_l \to V_{l+1}$ (or $\varphi(V_l) = V_{l+1}$), and in general form

$$\varphi^t : V_l \to V_{l+t}, \ t \in \{1, 2, \ldots, L-l\}. \tag{2}$$

The set $V_{l+t}$ is the image of $V_l$, and $V_l$ is the preimage of the set $V_{l+t}$.

The number of vertices (3) and edges (4) of $G_L = (V_L, E_L)$ equals

$$N = N(n, L) = |V_L| = n^L, \tag{3}$$

where $n = |W|$ is the number of vertices of $H$.

$$M = M(n, q, L) = |E_L| = q\left(1 + n + n^2 + \ldots + n^{L-1}\right) = q\left(n^L - 1\right)/(n-1), \tag{4}$$

where $q = |Q|$ is the number of edges of $H$.

Let us consider in the sequence $G_1, G_2, \ldots, G_l, \ldots G_L$ a pair $G_{l-1}, G_l$ for any $l \in \{2, 3, \ldots, L\}$. The set of edges of *rank l* is the set $E_l \backslash E_{l-1}$ of edges that appear at stage $l$. And the element $e \in \{E_l \backslash E_{l-1}\}$ is accordingly called an *edge of rank l*.

As mentioned earlier, for canonical prefractal $(n, q, L)$-graphs $G_L = (V_L, E_L)$, the simplified term prefractal graph $G_L$ is used.

### 2.5.1. The Adjacency of Old Edges of a Prefractal Graph

An important characteristic of a prefractal graph is the preservation of the adjacency of old edges during generation. In the general case, the process of generating a prefractal graph is characterized by random incident connections of old edges with seed vertices, which replace the ends of old edges. However, in the study of complicated structures, special cases of generating prefractal graphs are important. When the adjacency of all old edges or only old edges of the same rank is preserved. If the adjacency of old edges of only one rank $l$, $l \in \{1, 2, \ldots, L\}$ is preserved, then we say that the prefractal graph

$G_L$ is generated while preserving the adjacency of old edges of rank $l$. If the adjacency of old edges of all ranks is preserved, then the prefractal graph is generated with the adjacency of old edges preserved (see Figure 9a). In the event that the old edges of rank $l$, $l \in \{1, 2, \dots, L\}$ are not adjacent (do not intersect), then it says in the prefractal graph $G_L$ the old edges of rank $l$ do not intersect. If the old edges of any ranks do not intersect, then in the prefractal graph, the old edges do not intersect (see Figure 9b).

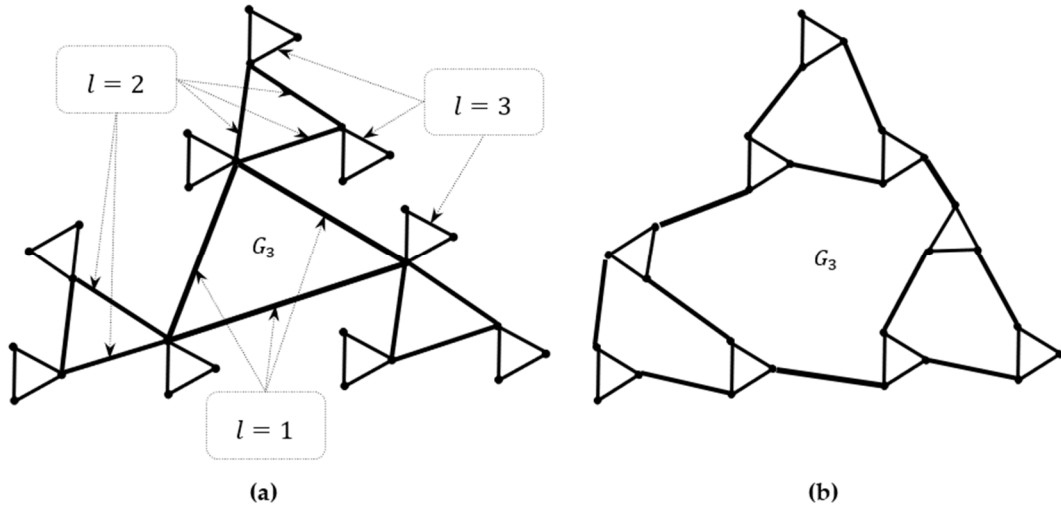

**Figure 9.** (**a**) Prefractal graph with preservation of adjacency of old edges. (**b**) Prefractal graph in which the old edges do not intersect.

Conditions for preserving or crossing old edges of certain ranks, for example, even or odd, are negotiated additionally. For example, a prefractal graph in which old edges of equal ranks do not intersect, while old edges of different ranks can intersect.

**Statement 1.** *Any prefractal graph $G_L$ generated by a single seed $H$ can be constructed with non-intersecting old edges.*

In this statement, non-intersecting edges means non-adjacent edges. The maximum number of edges $q = n(n-1)/2$ is present in the full seed $H$. Then, in accordance with the generating procedure for $G_L$ $N = n^L$, $M = \frac{q(n^L - 1)}{n - 1} = \frac{n(n-1)}{2} \cdot \frac{(n^L - 1)}{n - 1} = \frac{n(n^L - 1)}{2}$. At the first step, the graph $G_1 = H$ has $n$ vertices and $n(n-1)/2$ edges. Each of the $n$ vertices of the complete graph is incident with $(n-1)$ edges, and all edges are new. At the next step, to generate the graph $G_2$, each of the $n$ vertices of $G_1$ is replaced by a seed $H$. Each seed from $G_2$ contains $n$ vertices of $n(n-1)/2$ new edges and connects $(n-1)$ old edges. Since there are $n$ vertices in the seed, and the number of ends of old edges is $(n-1)$, then all old edges are incident to different vertices and are not adjacent (do not intersect). In this case, in each seed, there remains 1 «free» vertex, to which no old edge is incident. At the third step, to generate the graph $G_3$, each of the $n^2$ vertices of $G_2$ is replaced by a seed $H$. Each seed from $G_3$ contains $n$ vertices of $n(n-1)/2$ new edges and matches $(n-1)$ old edges of the second rank. As in the previous step, in each seed, there is 1 «free» vertex, which is not incident with any old edge. The number of new seeds $G_3$ is equal to the number of vertices $G_2$ at the previous step $n^2$. The number of old edges of the first rank is $\frac{n(n-1)}{2} = \frac{n^2 - n}{2} < n^2$, if we count the ends of the old edges as $n^2 - n < n^2$. That is, all old edges of the first rank are placed in such a way that they are not adjacent to any old edges of different ranks. Thus, in $G_3$, the old edges do not intersect and there are n «free» vertices left. The construction of graphs $G_l$ at all steps $l = 1, 2, \dots, L$ in this way allows you to save the conditions of non-intersection of old edges (old edges are not adjacent).

**Consequence 1.** *For each prefractal graph of the trajectory $G_1, G_2, \dots, G_L$, Statement 1 is true.*

2.5.2. Blocks of Prefractal Graph

If from $G_L$ generated by the $n$-vertex $H$, we successively remove all edges of ranks $l = 1, 2, \ldots, L - 1$, then the original graph splits into a set $\left\{ B_L^1 \right\}$. Component $B_L^1$ is isomorphic to $H$. The elements of the set $\left\{ B_L^1 \right\}$ are called blocks of the first rank ($r = 1$). Similarly, when removing edges of ranks $l = 1, 2, \ldots, L - 2$ from $G_L$, we obtain a set $\left\{ B_L^2 \right\}$ of the second rank. Generalizing, when removing all edges of ranks $l = 1, 2, \ldots, L - r$ from $G_L$, we obtain a set $\left\{ B_{L,i}^r \right\}$ of *blocks of the rank* $r$, $r \in \{1, 2, \ldots, L - 1\}$, and $i = 1, 2, \ldots, n^{L-r}$ is the ordinal number of the block. $B_L^1 \subseteq G_L$ of the first rank is *subgraph-seeds* of $G_L$. It is obvious that any block $B_L^r = \left( U_L^r, M_L^r \right)$ is a prefractal graph $B_r = (U_r, M_r)$ generated by a seed $H$.

Let us specify a number of details for the mapping $\varphi$. For any vertex $v_j \in V_l$, $j \in \left\{ 1, 2, \ldots, n^l \right\}$ of $G_l = (V_l, E_l)$ from the trajectory $G_1, G_2, \ldots, G_L$, the following is true:

$$\varphi^t \left( v_j \right) = B_{l+t,j}^t, \tag{5}$$

where $B_{l+t,j}^t = \left( U_{l+t,j}^t, M_{l+t,j}^t \right) \subseteq G_{l+t}$, $t \in \{1, 2, \ldots, L - l\}$.

Similarly, $\varphi^t \left( B_{l,i}^r \right) = B_{l+t,i}^{r+t}$, $r \in \{1, 2, \ldots, L - t\}$, $i \in \left\{ 1, 2, \ldots, n^{l-r} \right\}$.

Two blocks of $G_L$ are adjacent if there is an edge between them. The fact that the blocks of $G_L$ are adjacent if and only if their preimages from (5) are adjacent does not require proof.

On Figure 10, blocks of the first rank $B_{L,1}^1$ and $B_{L,2}^1$ are adjacent through the edge $e'$, blocks of the second rank $B_{L,1}^2$ and $B_{L,2}^2$ are adjacent through $e''$, and blocks of different ranks $B_{L,2}^1$ and $B_{L,2}^2$ are adjacent through the edge $e'''$. The prefractal graph $G_3$ is generated by a complete 5-vertex seed with old edge adjacency preserved.

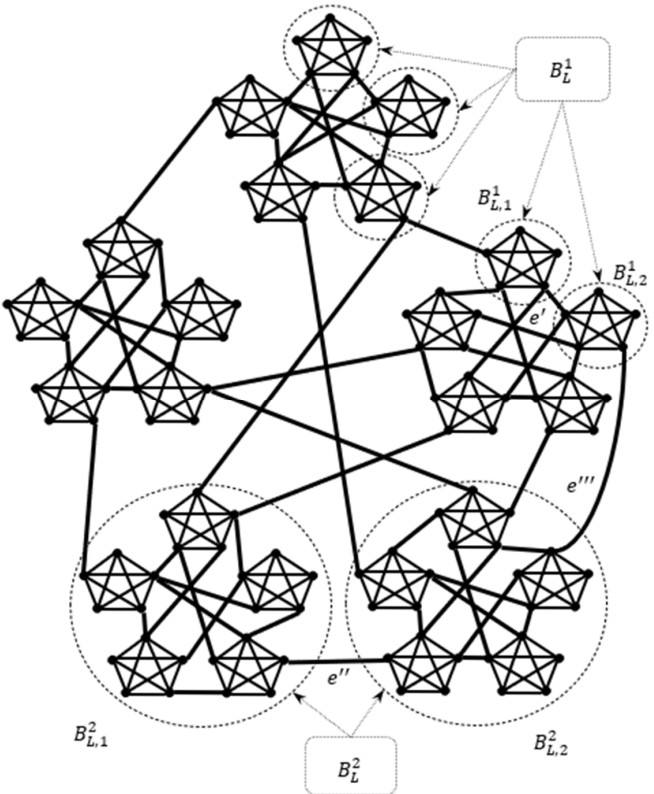

**Figure 10.** Blocks of prefractal graph $G_3$.

In what follows, the blocks of the prefractal graph are considered directly on the graphs without the procedure for removing old edges.

**Statement 2.** *Any prefractal graph can be represented as a set of subgraph-seeds connected by old edges of different ranks.*

Namely, the old edges of rank $(L-1)$ combine the set of subgraph-seeds $\left\{B_L^1\right\}$ into the set of blocks $\left\{B_L^2\right\}$ of the second rank, they, in turn, are combined by the old edges of the rank $(L-2)$ into the set of blocks $\left\{B_L^3\right\}$ of the third rank, and so on. Finally, the old edges of the first rank unite the set of blocks $\left\{B_L^{L-1}\right\}$ of the rank $(L-1)$ into a connected prefractal graph $G_L$. The process of connecting blocks of different ranks is shown in Figure 10.

### 2.5.3. Subgraph-Seeds of Prefractal Graph

The *subgraph-seed* $z_s^l$ is the block $B_{l,s}^1$, $s=1,2,\ldots,n^{l-1}$ of $G_l$, $l=1,2,\ldots,L$. Sequential selection of subgraph-seeds $z_s^l$ in the trajectory $G_1,G_2,\ldots,G_L$ splits the set of edges $E_L$ into non-intersecting subsets $Z(G_L)=\left\{z_s^l\right\}$, where $l=1,2,\ldots,L$, and $s=1,2,\ldots,n^{l-1}$. Such subsetting allows saving information about the adjacency of old edges at the time of their appearance in $G_L$. In what follows, for simplicity, the term *seed $z_s^l$ of rank l* is used.

The transition in the trajectory from $G_{l-1}$ to $G_l$ is carried out by $n^{l-1}=|V_{l-1}|$ RVS operations. Then the number of seeds of $G_L$ is $1+n+n^2+\ldots+n^{L-1}=\frac{n^L-1}{n-1}$. Then the cardinality $|Z(G_L)|=\frac{n^L-1}{n-1}$.

### 2.5.4. Weighting Prefractal Graphs

Consider the weighting of $G_L=(V_L,E_L)$ generated by $H=(W,Q)$ with $|W|=n$, $|Q|=q$. $G_L$ is weighted if its edges $e^{(l)}\in E_L$ is assigned a real number $w\left(e^{(l)}\right)\in\left(\theta^{l-1}a,\theta^{l-1}b\right)$, where $l=\overline{1,L}$, $a,b>0$, $a<b$ and $0<\theta<a/b$—similarity coefficient $\theta\in(0,1)$.

A vertex-weighted prefractal graph is defined similarly. $G_L$ is vertex-weighted if its vertices $v^{(l)}\in E_L$ is assigned a real number $v\left(e^{(l)}\right)\in\left(\theta^{l-1}a,\theta^{l-1}b\right)$, where $l=\overline{1,L}$, $a,b>0$, $a<b$ and $0<\theta<a/b$—similarity coefficient $\theta\in(0,1)$.

Without violating the rules for weighting a prefractal graph in the classical sense, we define the edge weights as follows. A prefractal graph $G_L$ is interval-weighted if each of its edges $e^{(l)}\in E_L$ is assigned an interval number $w\left(e^{(l)}\right)=[\underline{w},\overline{w}]\subseteq\left(\theta^{l-1}a,\theta^{l-1}b\right)$.

These definitions are also true for prefractal graphs generated by a set of seeds.

### 2.6. Some Theorems and Consequences for the Class of Prefractal Graphs

A graph is connected if it contains exactly one connected component. Which means that there is at least one path between any pair of vertices in this graph.

**Theorem 1.** *A prefractal graph (canonical or noncanonical) generated by a connected seed is connected.*

**Proof of Theorem 1.** If the prefractal graph is undirected, the existence of one connected component allows one to construct a path from one vertex to any other. The procedure for generating a prefractal graph guarantees that a connected prefractal graph will be obtained for a connected seed. □

**Consequence 2.** *A prefractal graph from the trajectory $G_1,G_2,\ldots,G_L$ (canonical or noncanonical) generated by a set of connected seeds is connected.*

A digraph is strongly connected if all its vertices are mutually reachable. That is, between any two vertices there is a path in both directions.

**Theorem 2.** *A prefractal digraph (canonical or noncanonical) generated by a strongly connected seed is strongly connected.*

**Proof of Theorem 2.** For a prefractal digraph, it is not always possible to find an oriented path from one of its vertices to another. Vertices can only exist with incoming or outgoing arcs. Consider the procedure for generating a prefractal digraph by a strongly connected seed. The prefractal digraph $G_1$ in the trajectory $G_1, G_2, \ldots, G_L$ is taken equal to the seed $H$, that is, each of its vertices is reachable from any other vertex. At the next step of constructing the digraph $G_2$, the vertices of $G_1$ are replaced by seeds. Let us consider this process in detail. We replace only one vertex $v_1$ of the digraph $G_1$ with the seed $H$. If the adjacency of the old arcs incident to the vertex $v_1$ is preserved, the seed is actually glued together at the common vertex $v_1$ (see Figure 11a,b).

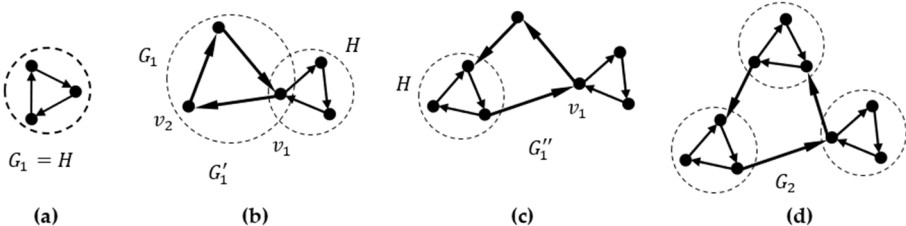

**Figure 11.** The procedure for generating a prefractal digraph $G_2$ with a 3-vertex complete oriented seed $H$ : (**a**) prefractal digraph seed $H$; (**b**) the seed is glued at the common vertex $v_1$; (**c**) replacing vertex $v_2$ with seed $H$; (**d**) prefractal digraph $G_2$.

In the constructed digraph $G_1'$, all $n$ vertices of the digraph $G_1$ are reachable among themselves, and $n$ vertices of the seed $H$ are also reachable. However, in the digraph $G_1'$, one vertex is common for $G_1$ and $H$. Then it is possible to build a path from any vertex $G_1$ to the common vertex $v_1$ and further from it to any vertex of the glued seed $H$. It is also possible to build a path from any vertex of the seed $H$ to a common vertex $v_1$ and then from it to any vertex of the subgraph $G_1$ of the digraph $G_1'$. That is, every vertex is reachable from every other vertex of $G_1'$. We replace the next vertex $v_2$ of the digraph $G_1$ with the seed $H$, and in fact of the digraph $G_1'$ (see Figure 11c). Suppose that the adjacency of the old arcs incident to the vertex $v_2$ is not preserved. Old arcs become incident to new seed vertices in an arbitrary order. However, the seed $H$ is a strongly connected graph, and then the ends of previously adjacent arcs are mutually reachable via the paths of the seed. We obtain that all vertices of the digraph $G_1''$ are mutually reachable. Replacing in turn the remaining $(n-1)$ vertices of the digraph $G_1$, we obtain a digraph $G_2$ in which each vertex is reachable from any other vertex, which means that the prefractal digraph $G_2$ is strongly connected (see Figure 11d). Replacing alternately the vertices of the digraph $G_2$, we construct a digraph $G_3$, which is also strongly connected. Using the above reasoning, we construct prefractal digraphs $G_1, G_2, \ldots, G_L$. At the $L$-th stage, we obtain a prefractal digraph $G_L$, which is strongly connected. For a noncanonical prefractal digraph, not all vertices are replaced by seeds during generation, that does not affect the proof of the theorem. $\quad\square$

**Consequence 3.** *A prefractal digraph from the trajectory $G_1, G_2, \ldots, G_L$ (canonical or noncanonical) generated by a set of strongly connected seeds is strongly connected.*

### 2.7. The Concept of a Fractal Graph

In accordance with the procedure for generating a prefractal graph, a fractal graph is determined. Let the trajectory $G_1, G_2, \ldots, G_l, \ldots, G_L$ be given. As $l \to \infty$, the graph $G_l$ is fractal. For a fixed $l$, a prefractal graph is considered. For $l = L$, the prefractal graph $G_L$ is considered.

It should be noted that the infinite sequence $G_1, G_2, \ldots, G_{l \to \infty}$ moves to the right starting from the first element $G_1$ and then the next one. That is, in such a sequence, you

can always select the first element $G_1$, the second $G_2$ and all subsequent ones. Each graph in this sequence is a finite prefractal graph.

Another kind of fractal graph is an infinite sequence in both directions $\ldots G_{l-1}, G_l, G_{l+1}, \ldots$. In this case, the fractal graph has no initial value $G_1 = H$ and each graph in the trajectory $\ldots G_{l_0-1}, G_{l_0}, G_{l_0+1}, \ldots$. is actually fixed by the observer. The graphs in the $\ldots G_{l_0-1}, G_{l_0}, G_{l_0+1}, \ldots$. sequence are infinite. Moreover, the vertices exist only at the time of fixing the graph $G_{l_0}$.

The following questions require further study: how the similarity coefficient affects the weights; at what values of the similarity coefficient there are limiting values of units or total weights; what the ratio will be of the cardinalities of the sets of vertices and edges in the trajectory, etc. It also requires the development of methods for searching for a set of generating seeds, adaptation of algorithms designed for prefractal graphs, calculation of structural, and numerical characteristics of infinite (fractal) graphs.

A fractal graph, for example, can be used as a tool for modeling large-scale clustering of matter in the universe. However, in practical applications, prefractal graphs, including large ones, are mainly used, which allow modeling artificial objects. A separate work will be devoted to the study of fractal graphs, including the definition of fractal dimension, and the possibility of saving the results obtained on prefractal graphs. Refinement of the theory will be made with correction for the infinity of graphs.

## 3. Results and Discussion

This paper proposes an introduction to the class of fractal (prefractal) graphs. The first mention of fractal graphs can be found in [44]. Since then, the terminology of fractal graphs has firmly established itself in graph science. Nevertheless, different authors, understanding and speaking about the same objects, both fractals and their models—fractal graphs—offered independent definitions and descriptions. In this paper, the authors propose a general description of fractal (prefractal) graphs as a class of graphs. It is worth mentioning that in this work, the main attention is paid to prefractal graphs, as finite analogues of fractal graphs. Speaking in a general sense about fractal graphs and about the class of fractal graphs, both types of finite and infinite graphs are meant. If necessary, for the rigor of statements, the term prefractal graph is used.

Thus, to determine prefractal graphs, the main operations for their generation are proposed. A fractal graph is known to have the property of self-similarity, where a part of the graph is similar to its other parts or to the entire graph. The minimal possible similar parts are called seeds. In the general literature, the concepts of graphlets [45–47] and motifs are used. Graphlets are often used in the analysis of the structure of graphs and, in fact, are equivalent to the concept of seeds in fractal graphs.

Motives are similar in meaning to graphlets but are used for the statistical analysis of large-scale networks, including the identification of properties and characteristics [48–51]. The types of motives and the number of their occurrences in the graph are calculated. In fact, the graph is divided into many subgraphs—motives. A set of algorithms for identifying motives and their further statistical processing is developed.

The operation RVS is considered in detail, when the current vertex is removed and a subgraph, a seed, is inserted instead. The procedure for generating a prefractal graph is a step-by-step replacement of all vertices with seeds. At the output of each stage, a prefractal graph is formed. The sequence of stages of generation of prefractal graphs forms a sequence $G_1, G_2, \ldots, G_L$ called a trajectory. If the rules for generating a prefractal graph are observed, we are talking about a canonical, that is, a typical graph. If the generation procedure is violated or additional restrictions are introduced, then one speaks of a noncanonical (special) graph. $G_L$ is also generated not by one, but by many seeds.

For a complete description of the class, directed graphs and multigraphs with multiple edges and loops are also briefly considered.

Special attention is paid to the description of the adjacency property of old edges, which affects the formation of the graph structure, including the generation of model

graphs. The concepts of blocks and seed subgraphs of different ranks are also considered. The ranks of individual subgraphs, edges or vertices indicate their belonging to a particular generation stage.

The rules for weighting prefractal graphs by real numbers and intervals are given. The issue of weighting fractal graphs requires separate consideration. It is also necessary to expand the concept of multi-weighting, including non-deterministic and fuzzy weights—time series, fuzzy sets, etc.

$G_L$ is also dynamic graph and represented by sequence $G_1, G_2, \ldots, G_L$ [26,28]. Formulations of multicriteria problems refer to a fixed $G_L$. It is required to consider the applicability of the statements to the sequence $G_1, G_2, \ldots, G_L$. It is necessary to consider the end-to-end work of algorithms and link the sequence of solutions.

At the end of the main part, some theorems and corollaries are given. Undoubtedly, it is required to expand the theoretical basis of the class of fractal graphs with a proof base, that is, to bring theorems with proofs. This will be done in the next publications.

Let us once again mention the difference between fractal (infinite) and prefractal (finite) graphs. In this paper, some descriptions of fractal graphs are given, and the main issues that require further study are identified.

The description of fractal graphs is intended to combine various families—self-similar graphs, Sierpiński graphs, scale-invariant graphs, etc. Flexible generation rules allow us to consider these and other well-known families of graphs with fractal properties. In this paper, in contrast to the works cited in the references, attention is paid to the rules of generation in topological time, where the trajectory shows how the graph developed. That is, the foundations for the study of fractal graphs as dynamic ones are laid. The above characteristics will allow us to continue the study of this class of graphs and take a fresh look at the problems of calculating dimensions, packing, genetic properties, etc.

## 4. Conclusions

This work is more of a theoretical nature; the main definitions, properties and characteristics of prefractal graphs are considered. In the future, there will be an expansion of research into the practical area, including for solving optimization problems. Prefractal graphs are used as a modeling tool (the structure of a social network [52,53], transport and logistics systems [54], processes in cryptocurrency networks [55,56], DNA structure, etc.).

Multiobjective optimization is applied on large graphs and complex networks [57–60], for example, in the problems of splitting a social graph [61] or transport and logistics problems [62,63]. The structure of prefractal graphs makes it possible to parallelize well-known sequential algorithms.

From an applied point of view, prefractal graphs are a universal tool for describing network interaction in multi-element network systems. For example, they are used to design the interaction structure of monitoring systems based on small UAVs [64]. In [65], we also have discussed six known NP-complete problems in relation to the class of prefractal graphs.

Thus, the class of prefractal (fractal) graphs has firmly taken its place in graph theory and has prospects for further research. We single out the following areas for further study of the class of fractal graphs:

(1) The study of seed properties, comparative analysis of graphlets and motifs; statistical analysis of prefractal graphs, identification of seeds (graphlets, motifs).

(2) The study of the properties of fractal (infinite) graphs, expansion of the conceptual base, description of the similarity coefficient, calculation of the fractal dimension; comparison of properties and characteristics of fractal and prefractal graphs; and studying the behavior of known and new algorithms on fractal graphs.

(3) The formulation of particular formulations of optimization problems on prefractal graphs, development of algorithms for finding solutions, building models, and practical applications.

(4)    The study of prefractal graphs as a subclass of dynamic graphs; introduction of terminology and theoretical basis; formulation of general problems; description of dynamic solutions; and the continuity and stability of solutions.

(5)    The study of well-known NP-complete problems (more than 300) in the class of prefractal graphs, formation of a general approach to identifying solvability conditions, etc.

In future works, it is planned to present new results in these areas, as well as to identify points of contact with network science and a number of other scientific sections.

**Author Contributions:** Funding acquisition, A.K.; Investigation, R.K.; Methodology, R.K.; Project administration, A.K.; Software, A.K.; Supervision, A.K.; Validation, R.K.; Visualization, R.K.; Writing—original draft, R.K.; Writing—review & editing, R.K. All authors have read and agreed to the published version of the manuscript.

**Funding:** This work has been supported by the grants the Russian Science Foundation, RSF No. 21-19-00481.

**Conflicts of Interest:** The authors declare no conflict of interest.

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
