# Peer review of "Introduction to the Class of Prefractal Graphs"

_mathematics, doi:10.3390/math10142500_

Round 1

Reviewer 1 Report

In this paper under review, the authors introduce a new class of pre-fractal graphs. The main definitions and notation are proposed – the concept of a seed, the operations of replacing a vertex with a seed (RVs), and the procedure for generating a pre-fractal graph. Canonical and non-canonical types of pre-fractal graphs are considered separately. Important characteristics are proposed and described – preservation of adjacency of edges in the process of generating a pre-fractal graph for different ranks in the trajectory. 

 This paper is the first work on the study of the properties and characteristics of fractal and pre-fractal graphs. I like the idea of investigating the fractal and pre-fractal graphs and I am happy to recommend the paper for publication in your journal. 

Author Response

Dear reviewer, thank you for the positive feedback and the opportunity to publish the article. In the future, I plan to continue research on this topic.

Sincerely, Rasul Kochkarov

Reviewer 2 Report

I read the author's contribution carefully and will give several suggestions. I'm optimistic and would like to see this paper published, but a minor revision is required. My concerns are listed below:

  1. The keywords must bean reduced to the five words without conjunctions. 
  2. In this sentence in the abstract, "Fractal graphs as a discrete representation are used to model and describe the structure of various objects and processes, both natural and artificial, like social networks." please write an add-in in which social network fractal graphs are used. In my opinion, in this way, your work will take an interest in interdisciplinary works.  
  3. Explain Statement 1 (page 11) and Statement 2 (page). Add the proof or give the references from where you get these Statements.
  4. In the Results and Discussion section, please explain your contribution to the field. How does your work differ from the works you cited in the references? Which are common points? 
  5. The conclusion is too long and defocuses the reader from the author's contribution to the field. The section in conclusion, " In one of the works of the authors [58], we propose an introduction to multicriteria discrete optimization for a class of prefractal graphs. ........." in my opinion, must bean transferred in the discussion section, and the authors must use active voice in this part. 

Author Response

Dear Reviewer,
responses to comments are given in the file.

Sincerely, Rasul Kochkarov

Reviewer 3 Report

The subject in the paper is still very relevant for research. The authors in an easy-to-read text introduce the concept of prefractal i.e. fractal graphs.  Starting with vertex splitting operation, they define the replace vertex with seed (RVS) operation and give a procedure for their construction.  By adjusting some details of RVS operation, different classes of prefractal graphs can be constructed.
Further, in the analysis some rather obvious properties of prefractal graphs are highlighted.
A more serious study of prefractal graphs introduced in this way is yet to take place. First more precise
definitions should be stated.
Despite the fact that there are no precise definitions, or even some deeper results, this paper provides a basis
for a more rigorous foundation of these concepts, which could be used to model and solve a wide
range of problems as mentioned in the conclusion.
This is the main result.

Author Response

Dear reviewer, thank you for the positive feedback and the opportunity to publish the article. I plan to continue my research on this topic in the future.

Yes, it is indeed necessary to move on to more precise definitions. I will try to further strengthen this part of the work.

Sincerely, Rasul Kochkarov

Reviewer 4 Report

The subject in the paper is still very actual for research. The authors in an easy-to-read text introduce the concept of prefractal i.e. fractal graphs. Starting with vertex-splitting operation, they define the replace-vertex-with-seed (RVS) operation and give a procedure for their construction. By adjusting some details of RVS operation, different classes of prefractal graphs can be constructed.

Further, in the analysis some rather obvious properties of prefractal graphs are highlighted. 

A more serious study of prefractal graphs introduced in this way is yet to take place. At first more precise definitions should be stated.

Despite the fact that there are no precise definitions, or even some deeper results, this paper provides basis for a more rigorous foundation of these concepts, which could be used to model and solve wide range of problems as mentioned in the conclusion.

Author Response

(The authors gave the same response as above.)
